# Artificial Intelligence and New Technologies in Melanoma Diagnosis: A Narrative Review

**DOI:** 10.3390/cancers17243896

**Published:** 2025-12-05

**Authors:** Sebastian Górecki, Aleksandra Tatka, James Brusey

**Affiliations:** 1Maria Sklodowska-Curie Medical Academy, Al. Solidarności 12, 03-411 Warszawa, Poland; 2Centre for Computational Science and Mathematical Modelling, Coventry University, Priory Street, Coventry CV1 5FB, West Midlands, UK

**Keywords:** melanoma, artificial intelligence, dermoscopy, reflectance confocal microscopy, optical coherence tomography, teledermatology, vision transformers, explainable AI, federated learning, clinical validation

## Abstract

Melanoma is one of the most dangerous skin cancers, and early detection offers the best chance for successful treatment. In recent years, digital technologies such as artificial intelligence and advanced imaging have begun to change how melanoma is detected and evaluated. Tools that analyze skin photographs, three-dimensional body scans, or microscopic images can help clinicians identify suspicious lesions more accurately and at earlier stages. This review provides an overview of how these technologies have advanced between 2020 and 2025, which systems are already in clinical use, and which remain under development. Important challenges are also highlighted, including inconsistent performance across different skin tones and the need for clear safety regulations. The summary aims to give non-specialist readers a clear understanding of how modern technologies may support earlier and more reliable melanoma diagnosis.

## 1. Introduction

Melanoma remains one of the most aggressive and diagnostically challenging skin cancers. Although it represents only 1% of all skin malignancies, it is responsible for most deaths related to skin cancer worldwide [1,2]. In 2022, GLOBOCAN reported 331,722 new melanoma cases and 58,667 deaths, the highest incidence observed in high-HDI regions such as Oceania, North America, and Europe, where fair-skinned populations face substantial ultraviolet (UV) exposure [1,2]. In the United States, the incidence of melanoma continues to increase by 2 to 3% annually, while mortality has decreased due to earlier detection and advances in targeted and immunotherapies [3]. Early diagnosis remains the strongest determinant of survival: localized melanoma has a five-year survival rate above 99%, compared to 30% for metastatic disease [4].

Despite overall progress, significant diagnostic inequities persist. Individuals with darker skin tones often present in later stages and experience worse outcomes; in the United States, five-year survival is 94% among White patients but only 70% among Black patients [5]. These disparities reflect long-standing demographic biases in medical imaging datasets and highlight the ethical imperative to prevent AI systems from perpetuating unequal performance [6,7].

Conventional melanoma diagnosis remains labor-intensive and depends on the expertise of the clinician. The initial evaluation is based on visual inspection supported by the ABCDE criteria and the “ugly duckling” sign [8]. Dermoscopy improves diagnostic accuracy, but remains subjective and operator-dependent [9]. Suspicious lesions require excision and histopathological confirmation, the gold standard for staging and management [10]. This multi-step workflow is resource-demanding and often inefficient; in many healthcare systems, the number of biopsies needed exceeds nine [11], a challenge exacerbated by workforce shortages in dermatology.The past five years (2020–2025) have marked a paradigm shift in melanoma diagnostics, driven by the maturation of AI and the expansion of teledermatology. This period is particularly significant. Around 2020, dermatologic AI entered a translational phase, driven by two key developments: the introduction of patient-contextual imaging in the ISIC 2020 Challenge and the rapid expansion of telemedicine during the COVID-19 pandemic [9]. Initially conceived as a supportive “second opinion” tool for specialists, AI has rapidly evolved into an active clinical augmenter capable of improving diagnostic accuracy and classification between levels of expertise [9]. Recent prospective studies demonstrate that AI-assisted decision support can increase non-specialist diagnostic performance by 10–15 percentage points [12] and reduce malignant lesions mismanagement from nearly 60% to less than 5% [12]. Recent developments in 2024–2025 further accelerated the translational trajectory of dermatologic AI. During this period, large multimodal foundation models emerged, capable of jointly analyzing dermoscopic, clinical, and even histopathological data. These advances enabled more robust cross-domain generalization [13] and shifted the field from isolated algorithmic benchmarks to clinically relevant performance.

Beyond conventional dermoscopy, AI is increasingly integrated with advanced non-invasive imaging modalities. Reflectance Confocal Microscopy (RCM) provides near-histologic in vivo cellular resolution, allowing AI to assist in margin assessment and reduce unnecessary biopsies [14]. High-Frequency Ultrasound (HFUS) and Optical Coherence Tomography (OCT) enable a quantitative evaluation of the depth and structure of the lesion, critical for preoperative planning and prognosis [15]. In addition, AI-supported 3D Total Body Photography (TBP) systems facilitate longitudinal monitoring and early detection of new or evolving lesions, a domain where algorithms can outperform human observers in change detection [16]. Together, these technologies are establishing a new diagnostic pathway that combines precision, scalability, and non-invasiveness. Currently, the field of AI has evolved technologically. The dominance of Convolutional Neural Networks (CNNs) has expanded to hybrid and Transformer-based architectures such as Vision Transformers (ViTs) and multimodal Foundation Models (FMs), which capture global image context and integrate clinical metadata [17]. Large-scale self-supervised and federated learning paradigms have enabled training on distributed, privacy-preserving data sources, thereby addressing demographic bias and improving generalizability.

These advances, together with key regulatory milestones, such as the U.S. Food and Drug Administration’s approval of the first AI-powered dermatologic device (DermaSensor, 2024) and the European Union Artificial Intelligence Act classifying medical AI as a high-risk category, illustrate the ongoing transition from experimental validation to real-world clinical implementation [18]. This narrative review synthesizes the developments in AI and digital technologies for the diagnosis of melanoma during the translation period of 2020–2025. It critically examines the evolution of algorithms, the quality and representativeness of dermatologic datasets, integration with advanced imaging modalities, and the progression toward clinical validation and regulatory approval. The objective is to bridge the gap between technical innovation and clinical application, highlighting both achievements and remaining challenges on the path to trustworthy, equitable, and scalable AI in dermatologic oncology. Several recent high-impact reviews have summarized developments in dermatologic artificial intelligence; however, most have examined either algorithmic innovation or clinical validation alone, without integrating the broader translational landscape. The present review adopts a more comprehensive perspective by synthesizing progress across four interdependent domains: (1) the algorithmic evolution from convolutional neural networks to vision transformers and multimodal foundation models; (2) the integration of AI with advanced non-invasive imaging modalities, including reflectance confocal microscopy (RCM), high-frequency ultrasound (HFUS), optical coherence tomography (OCT), and three-dimensional total-body photography; (3) the emergence of new regulatory frameworks—most notably the European Union Artificial Intelligence Act and the U.S. Food and Drug Administration’s Predetermined Change Control Plan (PCCP), which shape the development, deployment and oversight of adaptive dermatologic AI systems; and (4) translational barriers related to dataset heterogeneity, demographic underrepresentation, interoperability, and clinical workflow alignment. Unlike prior reviews, this work integrates methodological, clinical, regulatory, and translational perspectives into a single synthesis, emphasizing the 2020–2025 inflection period during which AI in melanoma transitioned from proof-of-concept systems toward regulated, clinically deployed tools. In contrast to earlier literature, our review includes a structured comparison of benchmark datasets, performance of leading architectures, prospective validation studies, regulatory obligations (FDA PCCP and EU AI Act), and emerging modalities such as hyperspectral imaging. By examining these dimensions collectively, this review situates current methodological advances within the broader translational inflection period of 2020–2025, during which the field shifted from proof-of-concept experimentation toward prospective validation and regulatory scrutiny. This integrated synthesis highlights persistent gaps in generalizability, fairness, and implementation science that remain insufficiently addressed in prior literature, and outlines priorities required to enable safe, equitable, and clinically scalable adoption of AI in melanoma diagnostics.

## 2. Materials and Methods

This article presents a narrative review of the scientific, technical, and regulatory literature concerning artificial intelligence (AI) and emerging imaging technologies in melanoma diagnosis, focusing on the translational period from 2020 to 2025. The year 2020 was selected as the starting point because it marks a turning phase in dermatologic AI research: the publication of landmark studies on human–AI collaboration [9], the introduction of contextual imaging in the ISIC 2020 Challenge, and the rapid adoption of teledermatology triggered by the COVID-19 pandemic. Collectively, these developments signaled a transition from proof-of-concept experimentation toward clinical validation and regulatory scrutiny. The review was designed to provide a comprehensive synthesis of cross-disciplinary progress that includes computer vision, dermatology, oncology, and medical device regulation, with an emphasis on transparency, reproducibility, and critical evaluation rather than quantitative meta-analysis.

### 2.1. Search Strategy and Data Sources

A structured literature search was conducted between November 2024 and October 2025 using major scientific databases and regulatory repositories. The databases included PubMed/MEDLINE, Scopus, ScienceDirect, IEEE Xplore, and Nature Portfolio, complemented by the *arXiv* preprint server to capture the latest advances in computer science and deep learning prior to peer-reviewed publication. Regulatory and policy documents were retrieved from the U.S. Food and Drug Administration (FDA), the European Commission (EU AI Act documentation), and related official portals to ensure comprehensive coverage of developments in AI-based Software as a Medical Device (SaMD). To enhance methodological rigor, an explicit screening process was implemented. After removal of duplicates, the titles and abstracts were independently reviewed by two reviewers. Full-text evaluation was conducted for all studies that met the inclusion criteria and disagreements were resolved by consensus. Although the present work is a narrative review, the search strategy and selection process were designed to emulate key elements of systematic methodology, thus improving transparency and reproducibility; full search logs and keyword combinations are available upon request.

### 2.2. Search Terms and Inclusion Criteria

Search queries combined medical and computational terminology using Boolean operators. The main keywords included “*melanoma diagnosis*”, “*artificial intelligence*”, “*deep learning*”, “*machine learning*”, “*convolutional neural network*”, “*CNN*”, “*vision transformer*”, “*ViT*”, “*reflectance confocal microscopy*”, “*RCM*”, “*high-frequency ultrasound*”, “*HFUS*”, “*optical coherence tomography*”, “*teledermatology*”, “*explainable AI*”, “*XAI*”, “*federated learning*”, “*EU AI Act*”, and “*FDA SaMD*”. Reference lists of key studies, systematic reviews, and meta-analyses were manually screened to identify additional relevant publications not captured by electronic searches. To ensure methodological rigor and consistency, studies were selected based on the following inclusion criteria:Original peer-reviewed research published between 1 January 2020 and 31 October 2025;Studies employing artificial intelligence or machine learning for the detection, classification, segmentation, or decision support of skin cancer or melanoma;Research integrating AI with advanced imaging modalities (e.g., dermoscopy, RCM, HFUS, OCT, or 3D total-body photography);Reports on prospective or real-world clinical validation, regulatory evaluation, or implementation of AI systems in healthcare workflows;English-language publications providing sufficient methodological detail to allow reproducibility and critical appraisal.

Exclusion criteria were defined as

Non-original works (e.g., editorials, commentaries, opinion pieces, or letters without primary data);Studies not related to melanoma or general dermatologic AI without clinical application;Purely algorithmic or technical research lacking medical or diagnostic validation;Duplicate analyzes of the same datasets without novel methodological or clinical insights.

### 2.3. Data Extraction and Categorization

The full text of all eligible studies was reviewed, and relevant information was extracted independently by multiple reviewers to ensure accuracy and consistency. Extracted data were organized into four major analytical domains:**Algorithmic Development:** model architectures (CNN, Transformer, hybrid, or multimodal), training datasets, image modalities, and key performance metrics (AUC, sensitivity, specificity, accuracy);**Data Resources:** characteristics of public and institutional datasets (e.g., HAM10000, BCN_20000, Fitzpatrick_17k, PAD-UFES-20), including sample size, histopathologic verification, and representation of Fitzpatrick skin types;**Clinical Validation:** study design (retrospective, prospective, or real-world), study population, clinical setting (dermatology, teledermatology, or primary care), and diagnostic endpoints;**Regulatory and Ethical Aspects:** adherence to AI reporting standards (TRIPOD-AI, STARD-AI, CLAIM), regulatory classification (FDA, EU AI Act), explainability, data privacy, and bias mitigation.

Because this review is narrative in nature, no formal scoring system such as QUADAS-2 was applied. Instead, emphasis was placed on identifying methodological trends, translational barriers, and convergence between technical innovation and clinical validation.

### 2.4. Scope and Limitations

This review does not constitute a systematic meta-analysis and therefore does not provide pooled quantitative estimates. Its qualitative synthesis focuses on conceptual and methodological evolution rather than on statistical aggregation of outcomes. The scope is confined to studies published between 2020 and 2025, corresponding to the emergence of transformer-based and multimodal foundation models, the first regulatory approvals of dermatologic AI devices, and the establishment of new legal frameworks such as the EU AI Act. Earlier studies were included only when foundational for contextual understanding. Several benchmark dermatology datasets incorporated in studies from 2020–2025 were originally released before 2020. Their inclusion reflects the fact that most contemporary AI research relies on longstanding public datasets for training and benchmarking, even when novel methodological contributions were published after 2020. Therefore, dataset release year should not be conflated with the publication year of the studies evaluated in this review. A total of 1246 records were screened during the initial search. After removing duplicates, 713 titles and abstracts were assessed for relevance, and 162 full-text publications were reviewed. Ultimately, 98 studies met all inclusion criteria and were incorporated into the narrative synthesis. A meta-analytic approach was not pursued because the included studies exhibit substantial heterogeneity in imaging modality, model architecture, outcome definitions, and evaluation metrics, precluding pooled quantitative analysis. Narrative synthesis was therefore deemed the most appropriate method to ensure accurate interpretation of methodological and translational trends. As a narrative review, this work is inherently prone to selection bias, even though structured search procedures were implemented. Unlike systematic reviews, narrative methodologies cannot fully eliminate author-driven selection and interpretation of evidence, which may influence the balance, emphasis, and comprehensiveness of the included literature. The authors acknowledge the inherent limitations of a narrative review, including potential selection bias, publication bias favoring positive findings, and the rapidly evolving nature of AI research, which may render some findings time-sensitive. Nonetheless, the chosen approach provides a comprehensive and critical overview of the current state of AI in melanoma diagnosis during its most dynamic phase of clinical translation.

## 3. Review of AI Advancements and New Technologies

The years 2020–2025 were marked by rapid progress in algorithmic architectures, data resources, and clinical applications of artificial intelligence (AI) in dermatology. This period represents a transition from proof-of-concept experimentation to prospective validation, regulatory evaluation, and early implementation in clinical workflows. The following sections summarize the principal technological, algorithmic, and methodological advances defining this translational era.

### 3.1. The Algorithmic Shift: From Convolution to Attention

#### 3.1.1. Consolidation of Convolutional Neural Networks (CNNs)

In the early 2020s, Convolutional Neural Networks (CNNs) consolidated their position as the dominant architecture for melanoma image classification [19]. Standard architectures such as ResNet, EfficientNet, Inception-v3, DenseNet, and MobileNet were widely benchmarked on reference datasets such as HAM10000 [20,21]. Many of these models achieved diagnostic accuracies exceeding 95% [22]. Despite these advances, CNNs remain limited by their local receptive fields and reduced ability to capture long-range dependencies across images. This limitation is critical in dermatology, where clinical judgment relies on holistic features such as lesion symmetry, color uniformity, and contextual comparison across body sites. Consequently, research focus shifted toward attention-based architectures capable of modeling global context.

#### 3.1.2. The Rise of Vision Transformers (ViTs)

Vision Transformers (ViTs) introduced the self-attention mechanism from natural language processing to computer vision, including medical imaging [23]. Specialized architectures such as DermViT and EViT-Dens169 employed hierarchical attention and multi-scale feature pyramids, achieving accuracies up to 97% on ISIC datasets while reducing parameter count by nearly 40% [24,25]. Hybrid architectures such as ConvNeXt and SkinSwinViT integrate convolutional feature extraction with Transformer-based contextual reasoning, achieving an optimal trade-off between accuracy and computational complexity [26,27]. Meta-analyses confirm the maturity of these architectures, reporting average AUC values between 0.96 and 0.98 in large-scale evaluations [28,29,30]. The models summarized in Table 1 represent the principal algorithmic paradigms that shaped melanoma classification research during the 2020–2025 period. Early convolutional neural networks (CNNs), including ResNet-50 and EfficientNet, established strong baselines across benchmark datasets such as HAM10000 and ISIC [19,20,21]. Their performance, although high, is constrained by the intrinsic locality of convolutional filters, which limits capture of global lesion structure. Multimodal fusion networks and foundation models further expand diagnostic capability by integrating dermoscopic imagery with contextual metadata, clinical photographs, or self-supervised representations. Examples include MILK10k-based multimodal systems and large-scale pre-trained models such as PanDerm and DermINO [31,32]. These models consistently demonstrate superior generalizability across institutions and patient populations, a trend confirmed by recent umbrella reviews and meta-analyses [28,29,30]. Overall, Table 1 illustrates the methodological progression from CNN baselines toward more expressive and scalable Transformer-based and multimodal architectures, highlighting the growing emphasis on generalization, contextual reasoning, and real-world clinical applicability.

The architectural transition from convolutional models toward attention-based and multimodal systems is summarized in Figure 1. Vision Transformers demonstrate superior performance over CNNs in melanoma classification for several reasons. Unlike CNNs, which impose fixed local inductive biases through convolutional kernels, ViTs learn spatial relationships dynamically. This flexibility allows them to adapt more effectively to heterogeneous imaging conditions and cross-device variability. Recent multimodal ViT architectures extend these capabilities further by integrating dermoscopic images with clinical metadata or textual descriptors. As a result, they enable context-aware diagnostic reasoning that more closely mirrors human clinical assessment. Nevertheless, ViTs require substantially larger pre-training datasets and higher computational resources. For this reason, hybrid CNN–ViT models remain an attractive compromise for real-time or resource-limited clinical environments. First, the self-attention mechanism provides a global receptive field, allowing the model to capture long-range spatial dependencies such as symmetry, border irregularity, and color variegation—features central to dermatologic diagnosis. Second, ViTs are more robust to variable lighting, sensor heterogeneity, and acquisition artifacts because of their patch-based image representation. Third, large-scale pre-training on non-dermatologic datasets improves generalization, particularly when fine-tuning on the relatively small datasets common in dermatology. Finally, ViTs naturally accommodate multimodal fusion with metadata or textual descriptors, making them well suited for clinically contextualized diagnostic tasks.

#### 3.1.3. Performance Benchmarks: AI Versus Dermatologists

Systematic reviews from 2023–2025 consistently demonstrate that modern AI systems perform comparably to experienced dermatologists in melanoma detection [28,29]. Pooled sensitivity and specificity values are approximately 86.3% and 78.4%, respectively, whereas generalist clinicians show markedly lower performance (sensitivity 64.6%, specificity 72.8%). These findings underscore the primary clinical value of AI as an augmentative tool for non-specialists rather than as a replacement for dermatologists. A landmark multicenter prospective trial conducted by Heinlein et al. (2024) further illustrated this translational inflection point [12]. The study demonstrated that an AI classifier achieved higher sensitivity (92.1%) than expert dermatologists (73.4%), but with a trade-off in specificity (67.3% vs. 82.8%). Combined human–AI evaluation achieved the best diagnostic balance (specificity 90.7%), confirming that collaborative decision-making yields the highest diagnostic reliability and reproducibility.

### 3.2. Data as the Foundation: Benchmark Datasets and Bias

Robust artificial intelligence (AI) systems in dermatology depend critically on the quality, scale, and diversity of the datasets used for training and validation [20]. Between 2020 and 2025, increasing scrutiny has highlighted several structural limitations of existing dermatology datasets, including incomplete biopsy confirmation, class imbalance, and substantial underrepresentation of darker skin tones [37]. These factors directly influence the external validity, fairness, and clinical safety of AI models. To contextualize the heterogeneity of available datasets, Figure 2 provides a schematic overview of the major public dermatology datasets categorized by imaging modality (dermoscopic, clinical, and multimodal). This visual map highlights the structural fragmentation of the dataset landscape, where dermoscopic datasets dominate algorithmic benchmarking, whereas clinical and multimodal datasets, despite their importance for real-world generalization remain substantially smaller and less standardized. A detailed comparison of these datasets—including imaging modality, sample size, biopsy-confirmation rate, and key limitations are provided in Table 2. Although several datasets listed in Table 2 were released before 2020, they remain the primary benchmark repositories used in nearly all dermatology AI studies published during the 2020–2025 period. Their inclusion reflects contemporary usage rather than dataset release chronology.

Table 2 provides a consolidated overview of existing and foundational public dermatology datasets relevant to AI research on melanoma. Section A includes core benchmark datasets that underpin most deep learning studies published between 2020 and 2025, while Section B contains supplementary and recent datasets that contribute additional imaging modalities (e.g., 3D total body photography), improved Fitzpatrick skin type diversity, or multimodal metadata (e.g., clinical context, patient-level variables). Although several datasets included in Table 2 were originally released before 2020, they remain the primary reference repositories used by nearly all contemporary AI studies of melanoma; therefore, they are included whenever they serve as essential training, validation, or external reference resources for research conducted during the period of 2020–2025.

The datasets summarized in Table 2 differ substantially in scale, imaging modality, representation of skin-of-color populations, and the proportion of biopsy-confirmed lesions. These differences have direct implications for model performance and clinical safety. For example, large-scale dermoscopic datasets such as HAM10000 and BCN_20000 provide strong baselines for algorithmic benchmarking but suffer from marked class imbalance and overrepresentation of lighter Fitzpatrick types, leading to documented performance disparities on darker skin tones [44,49]. Newer datasets such as DDI and SLICE-3D attempt to mitigate these limitations by providing broader skin type diversity or introducing novel imaging modalities. However, their scale remains relatively small, and most have not yet undergone prospective clinical validation.

The continued reliance on legacy datasets released prior to 2020 reflects their enduring role as standard benchmarks rather than an inconsistency in temporal scope. These datasets have been repeatedly repurposed in studies published between 2020–2025 due to their accessibility and well-established annotation protocols. However, this dependence reinforces structural biases: dataset origin (often a single geographic/clinical site), limited representation of acral and amelanotic melanoma, and heterogeneous annotation quality. Such factors contribute to decreased external generalizability, particularly when AI systems are evaluated on multiethnic or real-world datasets.

Addressing these limitations requires not only more demographically representative datasets but also methodological innovations such as domain adaptation, synthetic augmentation of underrepresented lesion types, fairness-aware optimization, and federated learning across multiple institutions. These approaches represent essential steps toward developing clinically robust and equitable AI systems capable of safe deployment in diverse patient populations.

#### 3.2.1. Algorithmic Bias and Representation of Skin of Color

Most public dermatology datasets remain dominated by lighter skin types, leading to systematic disparities in algorithmic performance across the Fitzpatrick scale [6,40]. The Fitzpatrick_17k dataset was created to quantify and mitigate this imbalance, while newer efforts such as the Diverse Dermatology Images (DDI) dataset directly addressed phototype diversity [44]. Recent reviews confirm that diagnostic accuracy of AI decreases significantly for darker skin tones, underscoring the ethical and clinical implications of biased datasets [49]. This issue has reached regulatory awareness: the U.S. FDA, in its 2024 De Novo authorization of the DermaSensor device, explicitly noted limited sensitivity data for Fitzpatrick IV–VI populations and advised clinical caution in these cases. Algorithmic fairness and representative data therefore remain central to future AI development.

##### Clinical Impact of Bias and Regulatory Responses

Performance disparities across Fitzpatrick skin types have direct clinical implications. Lower sensitivity in darker skin tones increases the risk of delayed melanoma detection, misclassification of atypical acral or amelanotic lesions, and deferred specialist referral. These disparities translate into measurable differences in stage at diagnosis and survival, exacerbating existing healthcare inequities. Regulatory bodies have begun to address this challenge. In the FDA’s 2024 De Novo authorization of DermaSensor, the agency explicitly required demographic subgroup analysis and issued cautionary guidance for Fitzpatrick IV–VI populations due to insufficient sensitivity data. Similarly, the EU AI Act mandates bias monitoring, representative training datasets, and transparency regarding demographic performance variation. Mitigation strategies extend beyond simply acquiring “more diverse data.” Effective approaches include domain adaptation, rebalancing techniques, synthetic augmentation of underrepresented subtypes, fairness constraints during model optimization, and federated training across institutions with heterogeneous populations. These methods represent an emerging technical framework for achieving equitable diagnostic performance.

### 3.3. Fusing AI with Advanced Imaging Modalities

In the 2020–2025 period, AI became increasingly integrated with advanced non-invasive imaging modalities, extending diagnostic capability beyond dermoscopy to subcellular and structural evaluation.

#### 3.3.1. Reflectance Confocal Microscopy (RCM)

RCM provides near-histologic, in vivo resolution of skin architecture. AI-assisted RCM algorithms have been developed for automated delineation of the dermal–epidermal junction and cellular morphology classification, demonstrating reductions in unnecessary biopsies exceeding 50% [14,50]. These systems routinely achieve AUC values near 0.97, positioning AI-enhanced RCM as a realistic step toward “digital biopsy” for equivocal lesions.

#### 3.3.2. Optical Coherence Tomography (OCT) and High-Frequency Ultrasound (HFUS)

AI-enhanced OCT and HFUS imaging allow non-invasive quantification of tumor depth and margin visualization, correlating strongly with histopathologic Breslow thickness (*r* = 0.88–0.94) [51]. These modalities show potential for preoperative staging and risk assessment, transitioning AI applications from purely diagnostic to prognostic use.

#### 3.3.3. Three-Dimensional Total Body Photography (3D TBP)

AI-supported analysis of serial 3D TBP imaging enables automated detection of new or evolving lesions in high-risk individuals. Early studies demonstrated that algorithmic longitudinal monitoring improves early melanoma detection rates by approximately 10% compared to manual review [16]. This approach establishes a paradigm for population-level surveillance and early intervention.

#### 3.3.4. Hyperspectral Imaging (HSI)

Hyperspectral imaging (HSI) has recently emerged as one of the most promising complementary modalities for early melanoma detection. Unlike dermoscopy or OCT, which analyze morphological structure, HSI captures reflectance spectra across tens to hundreds of narrow wavelength bands, enabling characterization of biochemical and microstructural properties of skin lesions that are invisible to RGB or dermoscopic imaging. Recent studies demonstrate that HSI combined with machine learning substantially improves discrimination between melanoma and benign lesions. For hyperspectral imaging, recent work has demonstrated that hyperspectral signatures enable robust separation of malignant and non-malignant lesions using both traditional machine learning classifiers and deep learning approaches, achieving high diagnostic performance across multiple skin cancer subtypes [52]. In a follow-up clinical evaluation, the Spectrum-Aided Vision Enhancer (SAVE) system achieved reliable detection of acral lentiginous melanoma, melanoma in situ, nodular melanoma, and superficial spreading melanoma, highlighting the modality’s sensitivity to early biochemical alterations preceding overt morphological change [53]. Across these studies, HSI models outperform RGB- or dermoscopy-based baselines, particularly for lesions with ambiguous morphology or in anatomically challenging locations. Importantly, the spectral dimension provides additional discriminatory power for acral and amelanotic melanomas, two subtypes that are frequently missed by both clinicians and RGB-based AI systems. Although promising, current HSI datasets remain relatively small and device-specific, and they lack widespread clinical standardization. Integration with deep learning–based spectral unmixing and multimodal fusion (e.g., combining HSI with dermoscopy or OCT) represents a critical next step for translating this modality into routine melanoma diagnostics. A consolidated comparison of the maturity and clinical evidence strength across major non-invasive imaging modalities is presented in Table 3.

### 3.4. The New Frontier: Multimodal and Foundation Models

#### 3.4.1. Radiomics and Multimodal Integration

Emerging research has increasingly sought to unify multiple data modalities—dermoscopic, histopathologic, and clinical metadata—to emulate human diagnostic reasoning [38]. Multimodal datasets such as MRA-MIDAS have enabled integrative modeling that links imaging and molecular features, supporting individualized risk stratification and precision diagnostics. These approaches bridge traditional image analysis with molecular dermatology.

#### 3.4.2. Self-Supervised and Foundation Models

The most recent phase of AI development (2024–2025) is characterized by the rise of general-purpose foundation models. Frameworks such as *PanDerm* and *DermINO* were pre-trained on millions of unlabeled dermatologic images using self-supervised and hybrid learning paradigms, demonstrating superior generalization across institutions and populations. When fine-tuned with limited labeled data, these models outperform conventional CNNs and Transformer-based systems. Foundation models also incorporate multimodal and vision–language learning principles. For example, CLIP-inspired architectures align visual features with dermatologic terminology, improving interpretability and transparency [32]. These developments represent a paradigm shift toward explainable, federated, and clinically scalable AI architectures capable of supporting a broad spectrum of dermatologic and oncologic tasks. The overall structure of an AI-assisted melanoma diagnostic workflow covering data acquisition across multiple imaging modalities, preprocessing and model inference, uncertainty estimation, clinician–AI collaborative decision-making, and integration with electronic health records is illustrated in Figure 3. This schematic provides a step-by-step overview of how AI systems operate within real-world dermatology workflows and highlights where human oversight is essential for safe clinical deployment. A comparative overview of the regulatory expectations under the U.S. FDA Predetermined Change Control Plan (PCCP) and the European Union Artificial Intelligence Act (AI Act) is presented in Figure 4. The figure contrasts core requirements for adaptive medical AI systems—including data governance, transparency, post-market monitoring, drift detection, performance auditing, and human oversight, while also identifying areas of convergence relevant to clinical translation. Together, Figure 3 and Figure 4 contextualize both the operational workflow and the regulatory landscape necessary for safe and scalable implementation of AI in melanoma diagnostics.

## 4. Clinical and Regulatory Translation

The transition of artificial intelligence (AI) in melanoma diagnosis from experimental research to clinical implementation between 2020 and 2025 has been accompanied by increasing emphasis on validation, safety, and regulatory compliance. This section reviews key clinical trials and emerging regulatory frameworks that define the current state of translational maturity in dermatologic AI.

### 4.1. Integration of AI into Clinical Workflows

Beyond standalone diagnostic accuracy, effective deployment of AI systems requires alignment with real-world clinical processes. Integrating AI into dermatology workflows involves several operational steps: (1) triage and risk stratification in primary care settings; (2) automated longitudinal monitoring through digital dermoscopy or total-body photography; (3) structured teledermatology pathways enabling asynchronous (store-and-forward) or synchronous review; and (4) interoperability with Electronic Health Records (EHR) via FHIR and DICOM standards. Successful AI integration depends on clinician trust, clarity regarding algorithmic uncertainty, and seamless incorporation into existing decision-making pathways. Cost considerations—including hardware requirements, workflow redesign, and reimbursement models—constitute additional barriers that influence adoption across healthcare systems. However, real-world integration remains challenging. Many dermatology clinics lack standardized imaging workflows, leading to variable image quality and inconsistent metadata capture. Furthermore, AI tools are often deployed in fragmented software ecosystems, where limited PACS–EHR interoperability creates additional barriers to routine clinical use.

### 4.2. Validated Clinical Applications

AI-based dermatologic devices have evolved from research prototypes to clinically validated tools authorized for real-world use. Prominent examples include the smartphone-based SkinVision application (CE-marked in the European Union) and the U.S. FDA-authorized DermaSensor device (De Novo classification, 2024). Both report sensitivities exceeding 90% in prospective studies; however, their real-world specificity remains a major limiting factor for clinical adoption [54,55]. The DermaSensor system represents a milestone in regulatory history as the first AI-enabled dermatologic device cleared for primary care physicians rather than dermatologists. Its diagnostic principle is based on elastic scattering spectroscopy (ESS) coupled with machine learning for lesion risk stratification. In the pivotal DERM-ASSESS III trial, the device achieved a sensitivity of 95.5% and a negative predictive value (NPV) of 98.1%, but at the expense of very low specificity (20.7–32.5%) [54]. This trade-off means that up to four out of five benign lesions flagged by clinicians as suspicious are still identified as “refer” by the device, underscoring the challenge of balancing overtriage with early detection.

Table 4 summarizes major clinical studies and regulatory outcomes from 2020–2025 that collectively reflect the evolving validation standards for AI-driven melanoma diagnostics.

Prospective studies collectively emphasize the necessity of independent, multi-institutional validation across diverse populations before AI systems can be trusted for widespread deployment. Moreover, these trials underscore that the greatest clinical value of AI lies not in replacement but in augmentation of clinician performance—particularly in primary and teledermatology settings where access to dermatologists is limited.

### 4.3. Regulatory Pathways: FDA and EU AI Act

The regulatory landscape for AI in medicine underwent major transformation between 2023 and 2025. In the United States, the Food and Drug Administration (FDA) finalized its *Predetermined Change Control Plan (PCCP)* in late 2024 to address the adaptive nature of AI algorithms. The PCCP allows manufacturers of AI-enabled Software as a Medical Device (SaMD) to predefine algorithmic modifications such as retraining on new data or recalibrating thresholds without requiring a new submission for each update, provided that safety and performance verification steps are explicitly specified.

In parallel, the European Union adopted the *Artificial Intelligence Act* (EU AI Act, 2024), which classifies AI-based medical diagnostic systems as “high-risk.” Under this designation, developers are required to implement comprehensive quality management systems, algorithmic transparency, data governance standards, and mechanisms for human oversight. Both frameworks represent pivotal progress but also highlight the remaining challenges for developers. Harmonization between regulatory jurisdictions, transparent validation protocols, and equitable performance across diverse populations will determine the pace and direction of AI adoption in clinical dermatology. A critical distinction should be made within both regulatory frameworks concerns: “locked” versus “adaptive” AI systems. Locked algorithms maintain fixed parameters post-deployment, whereas adaptive systems undergo continuous updates through periodic retraining. The FDA PCCP explicitly permits predefined algorithmic modifications, provided that manufacturers specify verification and validation steps, demographic subgroup performance analyses, and mechanisms for drift monitoring. The EU AI Act requires transparent logging, human oversight, and post-market surveillance, placing substantial operational responsibilities on developers. These requirements pose logistical challenges, particularly for small enterprises lacking resources for continuous monitoring, cybersecurity audits, and regulatory maintenance. Furthermore, unresolved issues surrounding liability—whether borne by physicians, institutions, or AI developers—remain an open barrier to widespread deployment.

#### Feasibility, Monitoring, and Unresolved Regulatory Gaps

Although the FDA PCCP and EU AI Act provide structured pathways for adaptive AI, their implementation presents major practical challenges. Continuous monitoring of model drift requires infrastructure for real-time performance auditing, demographic subgroup evaluation, and dataset version control—capabilities that may exceed the resources of small developers and non-academic institutions. In many clinical settings, routine recalibration is difficult due to limited IT capacity and inconsistent data streams. Liability remains an unresolved regulatory gap. As AI systems approach or exceed dermatologist-level performance, responsibility for diagnostic errors becomes increasingly ambiguous, with potential distribution across clinicians, institutions, manufacturers, and model maintainers. Clear legal frameworks defining performance thresholds and responsibility for diagnostic errors are still lacking. Addressing these gaps will be essential to enable safe, scalable deployment of adaptive dermatologic AI systems.

## 5. Discussion: Trust, Translation, and the Path Forward

### 5.1. Validated Technologies and Prospective Evidence

Validated AI technologies for the diagnosis of melanoma are those evaluated in prospective, multicenter, or real-world clinical settings. Examples include CNN- and Transformer-based dermoscopic classifiers tested in primary care [12,56] as well as CE- or FDA-authorized systems such as SkinVision and DermaSensor [54,55]. These tools demonstrate high sensitivity and clinically acceptable negative predictive value, although specificity remains a key limitation. Their evidence base is comparatively mature, but continued post-market evaluation is required to monitor performance decay, demographic subgroup disparities, and model drift.

### 5.2. Experimental Architectures and Early-Stage Research

Early-stage developments include foundation models, multimodal fusion networks, self-supervised learning frameworks, and hyperspectral imaging approaches. These architectures generally demonstrate better performance on retrospective test sets and cross-data set validation [32,38,39], but lack large-scale prospective clinical trials. Their generalizability remains limited by dataset biases, annotation variability, and institutional domain shift. As such, these models should be considered currently experimental and not directly translatable to clinical workflows.

### 5.3. Speculative and Emerging Future Directions

Speculative developments include diffusion models for rare-lesion augmentation, AI-driven prognostic modeling, wearable imaging technologies, and continuous monitoring platforms. Although conceptually promising, these ideas currently lack clinical validation, regulatory precedent, and implementation frameworks. Their translation will require new multimodal datasets, longitudinal study designs, and updated regulatory pathways capable of addressing dynamic, adaptive, and continuous learning systems.

### 5.4. Algorithmic and Technical Maturation

Between 2020 and 2025, artificial intelligence (AI) for melanoma diagnosis evolved from isolated proof-of-concept classifiers to multimodal clinically oriented diagnostic systems. Early achievements were driven by Convolutional Neural Networks (CNNs), which established strong baselines for image-based classification but were limited by local receptive fields. The subsequent rise of Transformer-based and hybrid architectures enabled modeling of global lesion context and inter-patient variability, marking a methodological turning point in dermatologic imaging. The introduction of Foundation Models (FMs) and Self-Supervised Learning (SSL) paradigms, exemplified by *PanDerm* and *DermINO*, demonstrated the feasibility of large-scale pre-training on millions of unlabeled dermatologic images followed by fine-tuning for specific diagnostic and prognostic tasks. These developments parallel trends in radiology and digital pathology, suggesting a convergence toward generalist medical vision models capable of reasoning across modalities rather than solving isolated tasks [23,57].

### 5.5. Explainable AI and Ethical Transparency

As algorithmic complexity increases, explainability has become central to the ethical and clinical acceptance of AI. Frameworks such as LIME, SHAP, and Grad-CAM enable clinicians to visualize which features contributed the most to the decision of a model, helping to distinguish genuine diagnostic cues from spurious correlations [58,59]. Beyond their interpretative role, explainability tools are now prerequisites for regulatory approval under both the U.S. FDA’s Software as a Medical Device (SaMD) framework and the European Union’s AI Act. Transparent and interpretable systems not only support regulatory compliance, but also improve clinician’ cognitive trust, a prerequisite for safe human–AI collaboration in medical decision-making [60,61].

### 5.6. Clinical Translation and Integration Challenges

The true clinical value of AI emerges not from standalone accuracy metrics, but from improved diagnostic workflows and patient outcomes. Prospective studies, including those in primary and teledermatology settings, demonstrate that AI assistance can increase sensitivity by approximately 10 to 13 percentage points and reduce mismanagement of malignant lesions from nearly 60% to below 5%. However, this performance gain is often achieved at the cost of reduced specificity, increasing the risk of overdiagnosis and unnecessary biopsies.

These trade-offs highlight the need for adaptive and context-aware systems capable of optimizing decision thresholds based on patient demographics and clinical setting [9,56]. A persistent translational barrier is technical interoperability: without seamless integration into Electronic Health Record (EHR) systems and Picture Archiving and Communication Systems (PACS), even highly accurate models remain underutilized.

Therefore, ensuring compliance with the DICOM and FHIR standards is essential to realize the full potential of AI in clinical workflows. Equally critical is clinician training, as understanding AI limitations and uncertainty estimates is fundamental to safe deployment [62].

### 5.7. Regulatory Evolution and Legal Liability

Regulatory adaptation has accelerated to keep up with the expanding use of adaptive algorithms in healthcare. The U.S. Food and Drug Administration (FDA) finalized its *Predetermined Change Control Plan* (PCCP) in 2024, allowing manufacturers to update AI models safely through predefined retraining procedures without repeating the entire approval process [63,64]. This framework marks a pivotal shift from static regulation to dynamic oversight, provided that each modification is documented, validated, and linked to performance monitoring.

In parallel, the European Union’s *AI Act* (2024) classifies AI-based diagnostic systems as “high-risk,” mandating robust risk management, data governance, transparency and human oversight [60]. Together, these frameworks signify the end of the unregulated era of medical AI and the beginning of a globally harmonized approach to algorithmic safety. Legal discourse increasingly suggests that, as AI systems achieve parity with expert performance, liability for diagnostic errors may shift from clinicians toward manufacturers and developers, emphasizing the need for shared accountability and robust post-market surveillance.

### 5.8. Privacy-Preserving and Federated Learning

Federated learning (FL) has emerged as a critical enabler of preserving privacy and facilitating equitable AI development. By allowing for distributed model training without direct data exchange, FL ensures compliance with GDPR and HIPAA while fostering multicenter collaboration [65].

Empirical evidence suggests that federated models often outperform their centrally trained counterparts in external datasets because they are exposed to more heterogeneous data distributions [66]. This not only improves robustness, but also mitigates algorithmic bias, promoting equitable diagnostic performance across diverse populations. As a result, FL represents both an ethical and a technical advancement in the pursuit of global-implementable AI for dermatology.

### 5.9. Limitations and Future Directions

Despite these advances, several structural limitations constrain the clinical adoption of AI in the diagnosis of melanoma. Most published studies remain retrospective, single-center, and based on homogeneous populations, limiting external validity [44]. Dataset bias continues to undermine performance on darker skin types, and few systems have undergone rigorous, prospective, multiethnic validation [13]. Future research should therefore prioritize.

**Prospective and demographically diverse trials** that assess the generalizability of the real world in healthcare systems and populations.**Standardized reporting frameworks** (e.g., TRIPOD-AI, CLAIM, STARD-AI) to ensure transparency, reproducibility, and comparability of results [67].**Explainable and federated architectures** that balance transparency with scalability and privacy preservation.**Multimodal foundation models** that integrate clinical, dermoscopic, histopathologic, and genomic data for the holistic characterization of melanoma [39].

Finally, continuous monitoring and calibration of AI systems deployed will be essential to maintain clinical safety and public trust. The path forward lies not in replacing clinicians, but in developing adaptive, equitable, and accountable AI systems that reinforce clinical judgment and extend diagnostic access to underserved populations.

## 6. Conclusions

Artificial intelligence (AI) for the diagnosis of melanoma has evolved from experimental classifiers to clinically validated and recognized regulatory systems. Between 2020 and 2025, the major advances were driven by the convergence of algorithmic innovation, data diversity, and clinical translation. CNNs matured into hybrid Transformer and foundation model architectures capable of multimodal reasoning, while the dermatologic community recognized that dataset bias—particularly underrepresentation of darker skin tones—remains a key obstacle to equitable performance. Integration with advanced imaging modalities such as Reflectance Confocal Microscopy (RCM), High-Frequency Ultrasound (HFUS), and 3D Total Body Photography (TBP) has extended the diagnostic reach beyond traditional dermoscopy.

Regulatory milestones, including the FDA’s Predetermined Change Control Plan (PCCP) and the EU AI Act, have formalized the accountability for adaptive AI systems and marked the beginning of global harmonization in the oversight of medical AI. Clinically, AI has transitioned from a diagnostic adjunct to a trusted augmentation tool, validated in multicenter prospective studies. The next phase of progress (2025–2030) will focus less on improving precision and more on achieving generalizability, transparency, and integration into healthcare ecosystems. Key priorities will include mitigating algorithmic bias, advancing explainability (XAI), deploying federated and privacy-preserving models, and defining liability frameworks that balance innovation with patient safety. Emerging technologies such as wearable microneedle biosensors illustrate the future trajectory toward continuous AI-driven disease monitoring. The fusion of biochemical detection and intelligent analytics could transform melanoma management from episodic screening to proactive real-time surveillance. If developed responsibly, these technologies will expand specialist expertise to underserved populations, enable earlier detection, and ultimately reduce global melanoma mortality. The task ahead is not to prove that AI can diagnose melanoma—but to ensure that it does so safely, ethically, and for everyone. A particularly promising direction is the integration of hyperspectral imaging (HSI) into future multimodal melanoma diagnostics. Unlike dermoscopy or OCT, which primarily captures the morphological structure, HSI provides dense spectral fingerprints that reflect the underlying biochemical composition. Recent studies have shown that HSI-based machine learning systems can accurately differentiate subtypes of melanoma—including acral lentiginous melanoma and melanoma in situ—using wavelength-dependent absorption and scattering patterns [52]. The Spectrum-Aided Vision Enhancer (SAVE) further demonstrated that spectral biomarkers may allow the detection of early carcinogenic changes before overt morphological alterations occur [53]. As multimodal foundation models mature, the incorporation of hyperspectral signatures alongside dermoscopic and clinical images can substantially improve early detection, especially for lesion subtypes that remain challenging for RGB-based AI systems or even experienced dermatologists. Additional emerging directions include the use of diffusion models for rare lesion enhancement, which can substantially improve performance in underrepresented melanoma subtypes. Large multimodal foundation models integrating clinical notes, dermoscopy, histopathology, and genomics represent a paradigm shift towards holistic patient-level reasoning. Self-supervised learning in unlabeled dermatology datasets offers a way to reduce the dependence on expensive expert annotation. Wearable diagnostic devices and smartphone-based spectroscopy can enable continuous monitoring in high-risk populations. Finally, AI systems are increasingly being developed not only for diagnosis but also for predicting future melanoma risk, enabling proactive surveillance strategies that reflect individualized risk profiles rather than episodic screening.

## Figures and Tables

**Figure 1 cancers-17-03896-f001:**
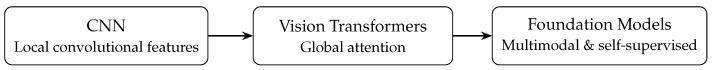
Evolution of deep learning architectures in dermatologic AI, progressing from convolutional neural networks (CNNs) to global-attention Vision Transformers, and finally to large-scale multimodal foundation models.

**Figure 2 cancers-17-03896-f002:**
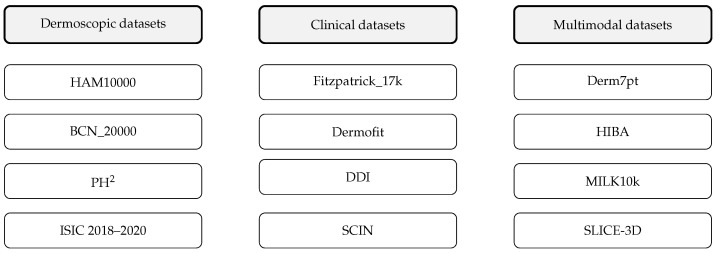
Schematic landscape of major public dermatology datasets categorized by imaging modality: dermoscopic datasets (**left**), clinical datasets (**center**), and multimodal datasets (**right**).

**Figure 3 cancers-17-03896-f003:**
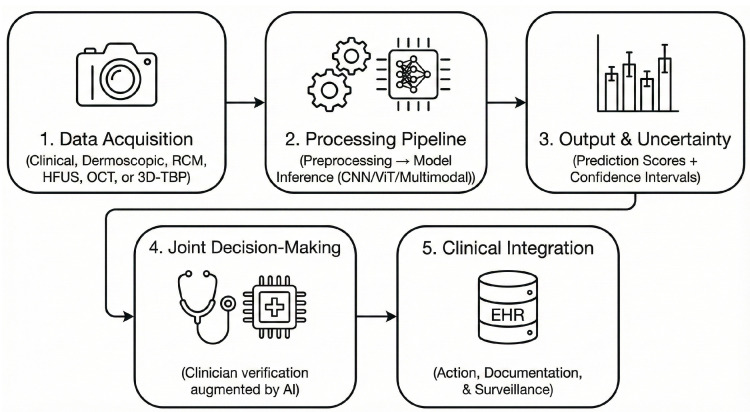
AI-assisted melanoma diagnostic workflow. The figure illustrates the full pipeline from image acquisition (clinical photography, dermoscopy, RCM, HFUS, OCT, or 3D-TBP), through preprocessing and model inference, to uncertainty estimation, clinician verification, and EHR integration.

**Figure 4 cancers-17-03896-f004:**
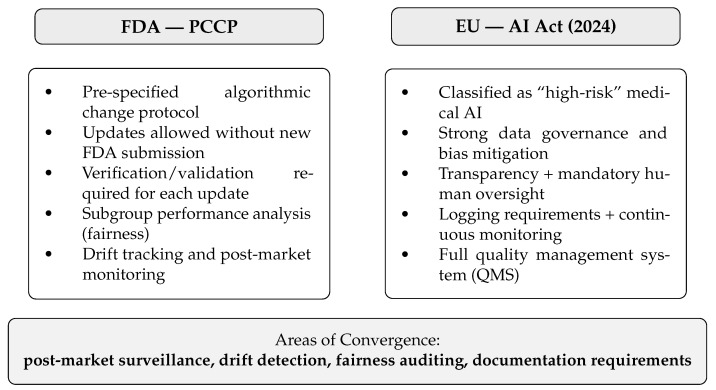
Comparison of the FDA Predetermined Change Control Plan (PCCP) and the EU Artificial Intelligence Act (AI Act). Each block summarizes regulatory expectations for adaptive AI in dermatology.

**Table 1 cancers-17-03896-t001:** Comparison of principal AI architectures for melanoma classification (2020–2025).

Model/Architecture	Dataset(s)	Input Modality	Performance (AUC)	Key Observations
ResNet-50/EfficientNet-B4	HAM10000 [20]; ISIC 2018–2020 [33,34,35]	Dermoscopy	0.94–0.96 [19,21]	Strong CNN baselines; limited global context [21]
DermViT/EViT-Dens169	ISIC 2019–2024 [34,36]	Dermoscopy	0.96–0.98 [24,25]	Superior global modeling; reduced parameter count [24,25]
Hybrid CNN–ViT (ConvNeXt, SkinSwinViT)	BCN_20000 [37]; MILK10k [31]	Dermoscopy + metadata	0.97–0.98 [26,27]	Combines local CNN features with global self-attention [26,27]
Multimodal Fusion Models	MRA-MIDAS [38]; MILK10k [31]	Dermoscopy + clinical metadata	0.95–0.98 [32,38]	Improved robustness and explainability through multimodal integration [32]
Foundation Models (PanDerm, DermINO)	Millions of unlabeled images + fine-tuning datasets [32,39]	Clinical + dermoscopy	0.97–0.99 [28,29,30]	Highest generalizability; strong external validation [28,29]

**Table 2 cancers-17-03896-t002:** Public dermatology datasets relevant to melanoma AI research (foundational and emerging).

Dataset	Size (Images)	Image Type	Biopsy Confirmation	Key Limitations
**Section A. Core Benchmark Datasets (commonly used 2020–2025)**				
HAM10000 [20]	10,015	Dermoscopic	>50%	Class imbalance; light-skin bias
BCN_20000 [37]	18,946	Dermoscopic	100% malignancies	Single-centre; limited FST diversity
Fitzpatrick_17k [6]	16,577	Clinical photos	Mixed	Skewed FST distribution; unbalanced
PAD-UFES-20 [40]	2298	Smartphone images	100% cancers	Small size; variable quality
PH^2^ [41]	200	Dermoscopic	20%	Very small; outdated
**Section B. Supplementary and Emerging Datasets (diversity, multimodality, 3D-TBP, metadata)**				
Derm7pt [42]	2000+	Dermoscopic + Clinical	Mixed	Limited diversity
Dermofit [43]	1300	Clinical	–	Restricted access
DDI [44]	656	Clinical photos	–	Small size; excellent FST diversity
ISIC 2018–2020 [33,34,35]	157,000+	Dermoscopic	Mixed	Heterogeneous annotation
SLICE-3D (ISIC 2024) [36]	400,000+	3D-TBP crops	Mixed	New modality; limited validation
Derm12345 [45]	12,345	Dermatoscopic	–	Limited accessibility
MILK10k [31]	5240 + 479 test	Multimodal (Clinical + Dermoscopy + Metadata)	–	Benchmark for multimodal models
HIBA Skin Lesions [46]	1616	Clinical + Dermoscopy	Mixed	Single-centre origin
SCIN [47]	10,000+	Clinical (skin, nail, hair)	–	Not melanoma-focused
SD-128/SD-260 [48]	6584	Clinical photos	–	Broad-spectrum dermatology

*Notes:* Section A lists core benchmark datasets underpinning the majority of melanoma deep learning research between 2020–2025. Section B includes datasets expanding imaging modality diversity (e.g., 3D total-body photography), Fitzpatrick skin type representation, or multimodal metadata availability. Dataset release year does not correspond to the publication year of AI studies; several pre-2020 datasets remain foundational for contemporary model development.

**Table 3 cancers-17-03896-t003:** Summary of maturity level, clinical evidence strength, and key considerations across major non-invasive imaging modalities used in melanoma diagnosis.

Modality	Maturity Level	Clinical Evidence Strength	Clinical Use-Cases/Limitations
Dermoscopy	High	Strong evidence from large retrospective datasets; supported by multiple prospective clinician–AI comparison trials.	Operator-dependent; image quality and device variability can affect performance.
RCM	Moderate	Several prospective trials showing high diagnostic accuracy and biopsy reduction.	Near-histologic resolution; limited availability, high cost, time-consuming acquisition.
OCT/HFUS	Moderate	Growing early-stage clinical validation; correlation with Breslow thickness frequently reported.	Useful for depth estimation and preoperative planning; limited resolution for subtle morphological changes.
3D TBP	Emerging	Limited but increasing prospective evidence; strong performance in longitudinal monitoring.	Best for high-risk patients; dependent on standardized image acquisition; emerging AI support for change detection.

**Table 4 cancers-17-03896-t004:** Selected clinical trials and regulatory outcomes relevant to AI-assisted melanoma diagnosis (2020–2025).

Device/Trial ID	Technology	Clinical Context/Regulatory Outcome	Sn/Sp for Melanoma
DermaSensor (NCT05126173)	Elastic scattering spectroscopy (ESS) with ML-based lesion risk classification.	Validated in the multicenter DERM-ASSESS III trial in primary care; authorized by the U.S. FDA under the De Novo pathway in 2024.	95.5%/ 20.7–32.5%
Dermalyzer (NCT05172232)	CNN-based decision support tool for suspicious lesion triage.	Evaluated in general practice; 2023 prospective study reported high diagnostic accuracy, supporting its use in non-specialist settings.	95%/86%
MoleMap AI (NCT04040114)	Deep learning classifier for melanoma detection and triage.	Assessed in dermatology specialist clinics; demonstrated substantial agreement with expert dermatologists and strong triage performance.	–
SkinVision	Smartphone-based CNN for self-screening and risk stratification.	CE-marked class IIa device used in teledermatology workflows across the EU; validated for high-risk lesion detection.	92.1%/80.1%

## Data Availability

Not applicable.

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
