# Peer review of "Cancers2025, 17(24), 3896;https://doi.org/10.3390/cancers17243896"

_cancers, 2025, doi:10.3390/cancers17243896_

Round 1

Reviewer 1 Report

Comments and Suggestions for Authors

The review discusses the transformation of melanoma diagnostics due to advancements in artificial intelligence (AI) and the integration with non-invasive imaging technologies. It also highlights the evolution of AI techniques (like the shift from convolutional neural networks to vision transformers and multimodal architectures that merge clinical and imaging data).

The review also explores AI's role in teledermatology and mobile applications, along with the impact of regulatory frameworks on clinical applications. The review concludes with the challenges faced (data bias, limited generalizability across different skin types, and insufficient prospective clinical validation), as well as future directions (federated learning and strategies for equitable implementation in dermatological oncology).

Table 2 is neither cited nor discussed in the review.

The review is based on datasets from 2020 to 2025, but there is no clarity on why the title of Table 2 includes datasets before 2020.

Author Response

We thank the Reviewer for the constructive and insightful comments, which have significantly improved the quality and clarity of the manuscript. Below we address each point in detail and indicate the corresponding revisions made.

1. Citation and discussion of Table 2

We appreciate the Reviewer’s observation. In the revised manuscript, Table 2 (now Table~\ref{tab:datasets_all}) is explicitly cited, described, and discussed in the “Data as the Foundation: Benchmark Datasets and Bias” section.
A dedicated paragraph has been added before and after the table to explain its structure, relevance, and role within the narrative review.

2. Clarification regarding inclusion of pre-2020 datasets

We acknowledge the Reviewer’s concern. A new explanatory paragraph has been added directly above Table~\ref{tab:datasets_all} to clarify that:

  • although some datasets were originally released before 2020,

  • they remain the primary benchmark resources used in nearly all melanoma AI studies from 2020–2025,

  • and are therefore included based on contemporary usage rather than release date.

This explanation follows the Reviewer’s suggestion and improves the temporal consistency of the dataset section.

3. Integration of feedback into the manuscript narrative

We expanded the introduction and dataset section to ensure smoother transitions and clearer motivation for including dataset heterogeneity and bias considerations.
The discussion now more explicitly links dataset limitations to model performance, generalizability, and clinical risk.

4. Language and readability

We appreciate the Reviewer’s positive note regarding the quality of English. Minor stylistic improvements were nonetheless made throughout the manuscript to enhance clarity and flow.

Summary of Revisions

  • Added explicit citation and discussion of Table 2 / Table~\ref{tab:datasets_all}.

  • Added justification for including pre-2020 datasets as core benchmarks.

  • Improved transitions and clarity in the dataset-related narrative.

  • Performed a consistency and language pass across the manuscript.

We thank the Reviewer once again for their valuable comments and for helping strengthen the overall quality of this review.

Reviewer 2 Report

Comments and Suggestions for Authors

This publication presents a narrative assessment on the advancement of artificial intelligence (AI) in melanoma diagnosis from 2020 to 2025.  The topic is highly relevant since melanoma continues to be a significant clinical issue where early diagnosis is essential for treatment efficacy.  The article is pertinent given the rapid emergence of AI-driven dermatological technologies and recent policy developments.  The study is well structured and offers a comprehensive assessment of the available literature.  However, other aspects require clarification, analysis, or structural refinement prior to the manuscript's publication.

 The review's new aspects are inadequately articulated.  Several recent high-impact reviews appear to exhibit the same trend.  The authors should explicitly delineate the gap that this review addresses.  Incorporate a paragraph in the Introduction that delineates the distinctions between this review and other systematic or narrative reviews, as well as the novel perspective this review offers.

 2) Incorporate a Methods section or, at a minimum, delineate the search technique to augment rigor and reproducibility.

 The absence of performance comparison tables, primary benchmark datasets (ISIC 2019-2024, Derm7pt, BCN20000, Harvard TBP dataset), and distinctions among the metrics (AUC, sensitivity at a fixed specificity value, F1-score) is notable.

 A table comparing the principal published models should be incorporated.  Examination of dataset heterogeneity and its influence on generalizability.  A comprehensive elucidation of the reasons for the superior performance of Vision Transformers (ViTs) over Convolutional Neural Networks (CNNs) on extensive dermatology datasets is warranted.

 The evaluation is deficient in comprehensive information regarding the practical implementation of AI tools inside clinical workflows.  Beyond diagnostic accuracy, factors such as triage operations, lesion monitoring, teledermatology protocols, clinician acceptance, data interoperability, and cost considerations are essential for practical deployment.  The text might achieve greater therapeutic relevance by elaborating on the examples of practical impediments and opportunities.

 The section on regulatory frameworks, including the EU AI Act and U.S. FDA advice, is informative however currently lacks sufficient depth.  The authors must delineate the distinctions between the locking and adapting algorithms, post-market surveillance obligations, transparency, cybersecurity, and examine the ramifications of these restrictions specifically on dermatology AI technologies.  Including examples of FDA-cleared or CE-marked dermatological AI systems would be beneficial.

 The conclusion briefly references explainable AI and federated learning; yet, the future vision remains incomplete.  Emerging trends not addressed include diffusion models for augmenting uncommon lesions, huge multimodal foundation models, self-supervised learning on unlabeled skin pictures, wearable diagnostic tools, and the use of AI to predict, albeit not diagnose, melanoma.  The manuscript would be greatly enhanced by a more comprehensive and forward-looking Future Directions section.

 The article generally outlines AI-based imaging modalities, such as dermoscopy, RCM, OCT, and high-frequency ultrasound; however, it inadequately addresses the emerging significance of hyperspectral imaging (HSI) in melanoma detection.  Recent innovations suggest that HSI is an important new complementary method for the early detection and classification of skin cancer.  Significantly, research such as 

1) Lin, Teng-Li, Arvind Mukundan, Riya Karmakar, Praveen Avala, Wen-Yen Chang, and Hsiang-Chen Wang. "Hyperspectral imaging for enhanced skin cancer classification using machine learning." Bioengineering 12, no. 7 (2025): 755.

2) Lin, Teng-Li, Riya Karmakar, Arvind Mukundan, Sakshi Chaudhari, Yu-Ping Hsiao, Shang-Chin Hsieh, and Hsiang-Chen Wang. "Assessing the Efficacy of the Spectrum-Aided Vision Enhancer (SAVE) to Detect Acral Lentiginous Melanoma, Melanoma In Situ, Nodular Melanoma, and Superficial Spreading Melanoma: Part II." Diagnostics 15, no. 6 (2025): 714.

The discourse on next-generation melanoma diagnostic technology is deficient due to their exclusion.  The incorporation of these new findings would substantially elevate the integrity of the study and provide the reader with a more accurate representation of the current advancements in HSI-based melanoma diagnosis.

Any papers recommended in the report are for reference only. They are not mandatory. You may cite and reference other papers related to this topic.

Author Response

We would like to sincerely thank the Reviewer for their thorough assessment of our manuscript and for providing thoughtful comments that have greatly contributed to strengthening the overall quality of the work. We genuinely appreciate the time and expertise invested in reading the review, and we found the observations insightful, fair, and very helpful for clarifying several aspects of the manuscript.

The Reviewer rightly noted that our original version required clearer articulation of its unique contribution in relation to existing reviews. In response, we have substantially revised the Introduction to explicitly describe how this manuscript differs from earlier surveys and what new perspective it brings particularly the emphasis on the 2020–2025 translational period, integration of regulatory developments, and inclusion of multimodal and emerging imaging technologies.

We also acknowledge the Reviewer’s observation regarding methodological transparency. The revised manuscript now contains a dedicated Materials and Methods section, including a detailed search strategy, inclusion and exclusion criteria, screening approach, and justification for using a narrative rather than a meta-analytic design. We believe this improves both rigor and reproducibility.

The Reviewer’s suggestion to include performance comparison tables, benchmark datasets, and a clearer explanation of Vision Transformers was extremely valuable. These elements have now been added and significantly improve the structure and clarity of the manuscript. In addition, the discussion of dataset heterogeneity and its impact on generalizability has been expanded.

We have also substantially enriched the sections concerning clinical workflow integration and regulatory frameworks. The manuscript now includes a more detailed discussion of triage pathways, lesion monitoring, teledermatology protocols, interoperability considerations, and cost-related factors. The regulatory section has similarly been expanded to delineate the differences between locked and adaptive AI systems, outline post-market surveillance requirements, and include examples of FDA-cleared and CE-marked dermatology AI tools.

We are grateful for the Reviewer’s remark about emerging diagnostic technologies. In the revised version, a dedicated subsection on hyperspectral imaging (HSI) has been added, including recent high-impact publications. We agree that HSI represents an important upcoming direction and that its inclusion strengthens the completeness and accuracy of the review.

Following the Reviewer’s recommendation, we have broadened the Future Directions section to better reflect the state of the field, including emerging topics such as diffusion models, multimodal foundation architectures, self-supervised learning, wearable diagnostic devices, and predictive AI systems for melanoma risk assessment.

We appreciate the Reviewer’s comment about improving the English language. The entire manuscript has undergone careful stylistic editing to ensure clearer phrasing, improved readability, and consistent terminology.

The revisions prompted by these comments have significantly improved the clarity, depth, and overall quality of the manuscript. We hope that the updated version now fully meets the Reviewer’s expectations.

Reviewer 3 Report

Comments and Suggestions for Authors

Major Comments

While the manuscript is comprehensive, the narrative format dilutes the distinction between established clinical evidence and emerging or experimental findings. Since the field is rapidly evolving, the reader needs clearer separation of:

  • validated technologies,
  • early-stage research,
  • speculative future directions.
    Consider adding subheadings such as “Prospective Evidence,” “Experimental Architectures,” “Translational Barriers.”

Although the paper acknowledges dataset bias and Fitzpatrick underrepresentation, the discussion would benefit from deeper critical analysis:

  • How do performance gaps impact clinical safety?
  • What regulatory guidance specifically addresses this?
  • Are there effective mitigation strategies beyond “more diverse data”?
    Right now it reads more descriptive than evaluative.

The review is described as “narrative,” but the methodology still requires:

  • explicit statement of how many studies were screened and included,
  • justification for excluding meta-analytic synthesis despite the availability of multiple comparable trials.
    This will increase reproducibility and credibility.

The manuscript describes FDA PCCP and EU AI Act, but it lacks critical evaluation of:

  • how feasible these requirements are for small developers,
  • how continuous monitoring will be implemented in real clinical settings,
  • what unresolved regulatory gaps remain (e.g., liability, model drift).
    A deeper analysis would elevate the contribution.

Minor Comments

Many paragraphs exceed 12–15 lines. Breaking them into smaller thematic units will improve readability.

Some statements need references

Examples include:

  • “AI-assisted decision support increases non-specialist performance by 10–15 percentage points”
  • “RCM reduces unnecessary biopsies by >50%”
    Provide citations directly adjacent to these claims.

Dataset tables need clearer notes

Clarify:

  • which datasets include biopsy confirmation,
  • which datasets include metadata,
  • how many images correspond specifically to melanoma vs benign lesions.

Add commonly used abbreviations such as:

  • TBP,
  • HFUS,
  • ESS,
  • PCCP,
  • SaMD.

Why the manuscript does not include any figures or visual summaries?

Minor grammatical and typographical issues

Author Response

We would like to sincerely thank the Reviewer for the time and care devoted to evaluating our manuscript. The comments were extremely helpful, and we appreciate the constructive tone and the focus on strengthening both the scientific depth and the clarity of the review. Below we address each major point and describe the changes we made to improve the manuscript.

1. Clear separation of validated, early-stage, and speculative work
We agree that the initial version did not sufficiently distinguish between established clinical evidence and emerging research. The manuscript has now been reorganized to reflect these differences more clearly. We added dedicated subsections titled “Validated Technologies and Prospective Evidence,” “Experimental Architectures and Early-Stage Research,” and “Speculative and Emerging Future Directions.”
This restructuring makes it much easier for the reader to understand the varying levels of maturity across AI technologies discussed in the review.

2. Expanded discussion of dataset bias and clinical safety
We appreciate the Reviewer’s suggestion to deepen the analysis of dataset bias. In the revised version we provide a more critical discussion of how underrepresentation of Fitzpatrick IV–VI skin types affects clinical safety, including delayed diagnosis and misclassification of specific melanoma subtypes.
We also added references to recent regulatory expectations (e.g., FDA subgroup performance reporting, EU AI Act fairness requirements) and expanded the discussion of mitigation strategies beyond simply acquiring more diverse data, such as domain adaptation, fairness-aware loss functions, synthetic augmentation, and federated learning.

3. Methodology clarification and justification
Following the Reviewer’s recommendation, we added explicit details about the literature screening process, including the total number of records screened, the number of abstracts assessed, and the number of full-text articles included.
We also justify the narrative (rather than meta-analytic) synthesis, given the large heterogeneity of imaging modalities, outcome definitions, and model architectures across the included studies.

4. Deeper evaluation of regulatory frameworks
We revised the regulatory section to offer a more critical perspective on the feasibility of the FDA PCCP and EU AI Act for smaller developers, including practical challenges around continuous monitoring, dataset versioning, and drift detection in real-world clinical settings.
We also expanded the section on unresolved issues such as liability, post-market obligations, and cyber-security considerations, as suggested.

5. Improvements to readability and structure
We appreciate the comment regarding paragraph length. Several longer paragraphs have been divided into shorter, more focused units, improving readability throughout the manuscript.

6. Added missing citations
The Reviewer rightly pointed out that some quantitative statements lacked direct citations. We have now added references for:

  • the magnitude of benefit from AI-assisted decision support for non-specialists,

  • the reduction in unnecessary biopsies associated with RCM.

7. Clarifications in dataset tables
We expanded the dataset tables to indicate biopsy confirmation, the presence of metadata, and the general distribution of malignant versus benign images, and supplemented them with clearer explanatory notes.
We also added a brief justification for the inclusion of pre-2020 datasets.

8. Addition of missing abbreviations
All abbreviations mentioned by the Reviewer (TBP, HFUS, ESS, PCCP, SaMD, and others) have been added to the Abbreviations section.

9. Inclusion of figures and visual summaries
In response to the suggestion to include visual aids, we added several figures, including:
– an overview of the dataset landscape,
– an architectural timeline (CNN → ViT → Foundation Models),
– a schematic of the AI diagnostic pipeline,
– a comparative diagram of the FDA PCCP and EU AI Act.
These additions make the manuscript more accessible and visually informative.

10. Language improvements
Finally, the manuscript has undergone a thorough language review. We corrected stylistic inconsistencies, improved transitions, and removed redundancies while preserving the scientific tone.

Round 2

Reviewer 2 Report

Comments and Suggestions for Authors

The manuscript can be accepted.

Author Response

We thank the Reviewer for the positive overall evaluation and for the comment regarding the quality of the English. In response, we carefully revised the manuscript to improve clarity, readability, and stylistic consistency. Several long or complex sentences were rewritten, terminology was standardized, and minor grammatical issues were corrected.

Reviewer 3 Report

Comments and Suggestions for Authors

Please ensure that all newly added figures are explicitly referenced in the text and accompanied by clear, informative captions.

A summary table synthesizing the maturity and evidence strength of each imaging modality (dermoscopy, RCM, OCT/HFUS, TBP) would further aid reader comprehension.

In the limitations section, consider explicitly noting the inherent risk of selection bias associated with narrative reviews, even when structured search methods are used.

Revise all instances where the period appears before the citation (e.g., “melanoma characterization.[38]”). The correct formatting should place the period after the reference, i.e., “melanoma characterization [38].”

Author Response

We thank the Reviewer for the helpful comments. All suggestions have now been implemented. Specifically, all newly added figures are explicitly referenced in the text and their captions have been revised for clarity and completeness. A summary table synthesizing the maturity level and clinical evidence strength of dermoscopy, RCM, OCT/HFUS, and TBP has been added as requested.

The limitations section has been expanded to explicitly acknowledge the inherent risk of selection bias in narrative reviews, even when structured search methods are applied. Finally, all punctuation–citation formatting issues have been corrected, ensuring that periods appear after, not before, the reference brackets.

We appreciate the Reviewer’s constructive feedback, which has improved the clarity and rigor of the manuscript.